# An Overview of Cryo-Scanning Electron Microscopy Techniques for Plant Imaging

**DOI:** 10.3390/plants11091113

**Published:** 2022-04-20

**Authors:** Raymond Wightman

**Affiliations:** The Sainsbury Laboratory, University of Cambridge, Bateman Street, Cambridge CB2 1LR, UK; raymond.wightman@slcu.cam.ac.uk

**Keywords:** cryo, scanning electron microscopy, sample preparation, hydration, plant–microbe interactions, cell wall, plant patterning, membranes, organelles

## Abstract

Many research questions require the study of plant morphology, in particular cells and tissues, as close to their native context as possible and without physical deformations from some preparatory chemical reagents or sample drying. Cryo-scanning electron microscopy (cryoSEM) involves rapid freezing and maintenance of the sample at an ultra-low temperature for detailed surface imaging by a scanning electron beam. The data are useful for exploring tissue/cell morphogenesis, plus an additional cryofracture/cryoplaning/milling step gives information on air and water spaces as well as subcellular ultrastructure. This review gives an overview from sample preparation through to imaging and a detailed account of how this has been applied across diverse areas of plant research. Future directions and improvements to the technique are discussed.

## 1. Introduction

The scanning electron microscope (SEM) is very much an interdisciplinary tool for characterising hard and soft materials in various ways. Depending on its configuration of detector(s), an SEM yields information on surface topology, roughness, morphology, compositional differences and sometimes sub-surface features. For biological applications, this translates to information about tissue organisation, the outlines of cells that make up these tissues, the surface profile of these cells and identification of sub-cellular features. The distinct types of information yielded by the SEM and transmission electron microscope (TEM) are becoming increasingly blurred, with embedded samples prepared with TEM workflows often used in the SEM for techniques centered around block-face imaging to observe the stained compartments. Some higher-end microscopes both “read” information of a sample and “write” to it, in terms of nanopatterning for generating new features on the samples.

The scanning electron beam requires a vacuum or partial vacuum depending on the level of imaging and, therefore, the specimen may need to be stable within this environment. For a plant specimen, as a form of biological material, this may require some preparation beforehand. Controlled dehydration that better preserves overall structure may be required (e.g., critical point drying, CPD). If the information is purely topological, then casts of the plant surface can be made and coated with conductive material to improve contrast. Casts are generally non-invasive and several casts taken at time intervals allow for growth-related observations, a time-course experiment, on the same specimen [1,2]. For some samples, freshly harvested tissue can be mounted directly within the SEM with no prior treatments and kept stable in either Variable Pressure (VP) or Extended Pressure (EP) mode, relying on low vacuum and sometimes a humid environment. Regardless of the method used, the aim is to observe and document the specimen as close to its natural state as possible, making every effort to limit or eliminate shrinkage. In this review, I present the problems associated with imaging fresh, hydrated tissue and current solutions, with a particular focus on cryo-SEM.

## 2. The Challenge of Imaging Hydrated Tissue in a Vacuum

For best performance in SEM imaging, in terms of resolving power and contrast, we need a good vacuum in order to get the best electron beam properties. This quickly dries the sample, resulting in considerable and rapid distortion. SEM instruments commonly support VP mode, resulting in user-defined target chamber pressures typically up to 133 Pa. VP appears to work very well for some tissue types (see Section 3) that normally tolerate some degree of dry environment, such as leaves with a thick cuticle, but appears to work poorly for tissue(s) that readily dry out at room temperature and atmospheric pressure, such as roots. EP or humid environmental imaging (pressures typically up to 1330 Pa) greatly improves the chamber conditions for normally damp samples (e.g., roots); however, reducing the vacuum in practice greatly reduces the level of detail and contrast.

Another challenge associated with a fresh, hydrated sample is a phenomenon known as charging. As this sample type is not normally conductive, it is prone to beam charging and associated damage of the sample. While a partial vacuum permits charging to be dissipated by the gas particles, the same particles hinder the beam properties. Coated CPD samples and casts are conductive and therefore overcome this issue. For fresh plant material, an evenly distributed conductive film would be required to be applied under vacuum, to ensure no subsequent tissue distortion. Alternatively, charging may be overcome by using low beam energies (commonly sub 1 kV); however, this may also alter the level of surface detail that is acquired. A useful resource on selecting the optimum accelerating voltage on an SEM has been put together by Dusevich et al. [3].

## 3. Imaging Cell Outlines Using Variable Pressure SEM of Fresh Hydrated Leaf Samples

Low-vacuum SEM imaging, focusing upon VP-SEM, is the subject of an excellent written overview and technical resource by Griffin [4]. For VP-SEM imaging as applied to plants, Talbot and White have optimised VP-SEM parameters that work well for imaging leaf epidermal cells [5]. Despite the best contrast being achieved for CPD-treated samples, the authors show that cell outlines are easily discernable using the backscattered electron detector (BSD, described in reference [5]) in VP mode, where fresh and hydrated leaf samples are stabilised by a moderately low temperature stage. The lower temperature reduces water loss from the leaf sample, giving up to 20 min useful imaging time. The use of VP-SEM for observing leaf epidermal cell morphology can be done with zero processing of the sample and, when compared to imaging cell outlines using confocal laser scanning microscopy (CLSF), requires no uptake of appropriate fluorescent dyes or transgenic fluorescent plants. This makes VP-BSD imaging in the SEM particularly appropriate for non-model plants. It also comes at a comparatively low cost since an expensive Field-Emission (FE-) SEM is not necessary to get good VP data, where only cell-level detail is required. Secondary electron imaging, using a variable pressure secondary electron (VPSE) detector on a FE-SEM, has been optimised to give cell surface level of detail for fresh, unprocessed, leaf trichomes for both glandular and non-glandular types [6]. Moreover, for the plant species that was studied with this setup, sample cooling is not required either before or during imaging [6]. As an extension of this technique, elemental mapping of trichome exudates has been demonstrated using energy-dispersive X-ray (EDX) microanalysis in VP mode [7].

## 4. An Overview of Low-Temperature SEM, Commonly Known as cryoSEM

CryoSEM involves the preparation and maintenance of fresh, hydrated material at cryogenic temperatures. Frozen material is suitable for imaging at high vacuum. It requires processing and additional equipment for freezing, fracture, sublimation, coating and transfer to the SEM chamber plus a cryo-stage within the chamber to keep the sample frozen during imaging. In our facility, the time between harvesting a plant sample and taking the first image under cryo conditions is in the order of 25 min and requires training and practice to successfully carry out the workflow. Getting the system at the required low temperatures takes around an additional 1 h. This is longer and requires more effort than imaging in a VP-SEM but, in our experience, cryoSEM is no more effort than optimising wet imaging for EP-SEM. Figure 1 shows a comparison between VP-SEM, EP-SEM and cryoSEM for the dissected Arabidopsis thaliana shoot apex, a challenging type of sample that is very sensitive to damage and drying. Only EP- and cryo-modes preserve organ- and tissue-level morphology for this type of sample.

The cryo workflow can be summarised as (i) Mounting the sample and placing in the correct orientation on a stub. Typically, the sample is mounted in Optimal Cutting Temperature (OCT) compound on a stub made of brass. (ii) Rapid freezing of the sample. (iii) Transfer under vacuum or inert gas to the SEM chamber with optional steps of cryofracture or cryo-ultramicrotomy, sublimation and conductive coating. (iv) Maintain cryo-environment in vacuum in order to prevent water loss and prevent exogenous ice forming. There are several commercial products, compatible with a variety of SEM brands and models, that incorporate all the steps from sample freezing to the cryo-stage in the SEM chamber ready for imaging. A diagram with still shots of one type of procedure is shown in Figure 2.

### 4.1. Sample Freezing

The freezing parameters are key to successful cryoSEM. They need to have ultra-low temperatures that freeze rapidly and then are maintained below a threshold for the remainder of the preparation and imaging. Optimal parameters keep the sample hydrated, limit ice damage and maintain the sample ultrastructure. The most commonly used cryogens are slushed nitrogen (slush N_2_) or liquid ethane and methods include plunge-, spray-, contact- and high-pressure-freezing. For a good comparison of these various methods and cryogens, you can refer to Echlin [8]. Although the cryogen needs to be appropriately very cold, it is not as simple as “the colder the better”. A key characteristic is the heat transfer and the ability to avoid significant film boiling, resulting in the production of an insulating layer of cryogen vapour that prolongs the freezing process—termed the Leidenfrost effect. To avoid or limit this effect, ethane is used as the preferred cryoprotectant; however, safety implications often make slush N_2_ plunging more desirable. This raises the question of why use slush N_2_ over liquid N_2_? The latter boils rapidly, resulting in a significant Leidenfrost effect, while the solid particulates in the slush firstly melt and are believed to be key in limiting the generation of any insulating layer. This is proposed to improve cooling rates up to 2-fold for samples in slush N_2_ [9].

The “holy grail” for cooling is to achieve vitrification, to rapidly freeze water to a glass-like amorphous material rather than cell-damaging ice crystals. A more accurate definition is the transition of water molecules from liquid to solid, coupled with no or very little change in their molecular configuration. The desired outcome has the frozen specimen being structurally identical to when its water was in a liquid state. Vitrification is difficult to achieve, especially as sample thickness increases. The process of “near-vitrification” results in nanocrystalline ice that may be sufficient for some applications [10,11].

### 4.2. Manipulations of the Frozen Sample

It is important to keep the sample frozen and to transfer it under vacuum or dry inert gas, since exposure to the atmosphere would result in rapid exogenous surface ice. Transfer between components of the equipment can be achieved by either maintaining the sample under liquid nitrogen or within a sealed vessel. As the sample is frozen, breaking open the cells allows for imaging of their contents, including membranes/organelles, cytoplasm and vacuoles. This can be achieved on a whole sample (e.g., stem, leaf) in various ways and includes (i) a cold knife that hits the ultra-cold sample creating a fracture, the location of which can only be controlled to a minimal degree, (ii) cryo-planing using a cryo-ultramicrotome [12] and (iii) cryo-planing using broad-ion-beam (BIB) milling [13]. Cryo-fractured tissue typically has complex topography and the fracture lines are sometimes influenced by structural/mechanical properties giving useful information on natural boundaries both at the tissue and subcellular scales (Figure 3a). Cryo-planing, on the other hand, allows for a greater level of precision and results in a relatively flat surface that is more suitable for downstream Energy Dispersive X-ray (EDX) analysis.

Contrast to resolve subcellular features is given by surface topology and/or compositional differences on the imaged surface. Imaging a fractured/planed cell is akin to imaging a block of ice, making intracellular membranes and other structures challenging to observe given that they are not always raised from the surface. This can be improved through an optional sublimation step (Figure 3b). Sublimation involves the removal of water from a frozen sample without going through an intermediate liquid phase and serves two purposes: (i) removal of exogenous ice during cryo preparation, (ii) removal of endogenous ice after cryofracture. For the latter, it means that the structure of the sample is maintained while removing surface ice, revealing subcellular structures that were not easily resolved prior to sublimation. This technique is also referred to as “freeze etching”. In practice, this involves raising the temperature from ultra-low temperature to low temperature for a duration of minutes. The longer the sublimation, the more the water surface retreats into the sample. An added benefit of sublimation appears to be better signal for elemental analysis [14].

### 4.3. Thin Layer Coating

A frozen biological specimen is essentially a non-conductive material. SEM imaging of such a specimen gives poor signal, can give rise to specimen charging and is susceptible to beam damage. Low beam energies or beam deceleration techniques can, to some degree, circumvent these issues but may not yield images with sufficient detail or contrast. Coating the sample with a thin conductive layer greatly improves the imaging. For cryoSEM, sputter coating is the preferred method of deposition and the conductive layer needs to be applied while the sample remains frozen. The cryo-stub with frozen sample represents the substrate and the sputtered material is ejected from the “target”, which is commonly a thin metal disk. The choice of sputtering material and the thickness of the conductive layer (typically between 1–30 nm thick) is dependent upon the desired signal and detail for SEM imaging. For low-magnification imaging (for plants this might be resolving cell outlines across a tissue), the grain size of the sputtered material is not as important as for high-magnification (e.g., small subcellular scale) SEM imaging where grain size becomes an important factor. It must be remembered that, at least for secondary electron imaging, the output on screen represents the surface of the conductive film and not the specimen itself and requires some consideration when choosing the target. A very useful resource is the comparisons between metal targets listed in Heu et al. [15] that includes grain sizes, desired magnification for imaging and sputter rates. Typical targets, listed in order of larger to smaller grain size, are gold (Au), gold/palladium alloy (Au/Pd), platinum (Pt), iridium (Ir) and chromium (Cr). In our facility, Au/Pd at a film thickness of 5–10 nm is used for most samples at magnifications up to a field-of-view that encompasses an entire plant cell. For imaging small subcellular regions such as plant cell wall ultrastructure, Pt or Ir (1–3 nm thickness) coatings are used.

### 4.4. SEM Imaging on the Cryo Stage

The vast majority of imaging follows the manufacturer’s recommendations for conductive samples at high vacuum. In the absence of thin layer coating (e.g., to monitor sublimation on the SEM cryo-stage in real time), imaging conditions for beam-sensitive materials will apply. While most work will involve detection of secondary electrons, backscattered electron detectors are good for easily distinguishing plant compartments and boundaries within a cell/tissue and, at high kV, thin metal coating does not seem to be detrimental to observing differences in atomic number (Figure 3c). For fractured droplets of a protoplast suspension, the wall-free cells can be challenging to locate using solely SE imaging but made much easier to find using the BSD detector. Cryo-maintained plant samples are amenable to X-ray microanalysis and the EDX technique has been successfully used to quantify or map the elements P, Ca, O in leaves and O, C, K, Ca, Cl and Na in roots [16,17]. For a detailed overview of this cryo-EDX technique, one may refer to McCully et al. [18].

Some high-end FE-SEM instruments can be fitted with a focused ion beam (FIB) column. The ions are many orders of magnitude heavier than electrons and can be used to eliminate material (milling) from a surface in the SEM chamber. The real power of the technique is precision regulation of the ion beam to remove defined small quantities of material, producing a new imaging face for subsequent SEM imaging. Iterations of milling and imaging provide 3D volume information (which is sometimes automated) and have gained new types of information from stained EM blocks—sometimes referred to as serial block-face imaging. FIB is not covered in detail here, and the reader is pointed to several good reviews [19,20]. For cryoSEM, FIB milling and in-chamber micromanipulator apparatus are gaining popularity for targeting and producing thin lamellae for transfer to a cryoTEM with ongoing improvements in speed and automation [21]. Omitting the TEM and carrying out all iterations of milling and imaging in the SEM chamber allows for rapid 3D volume imaging of the cryo-preserved sample [22], although it is important to be aware of FIB-induced artefacts [23]. A further use for the ion beam is imaging itself. He-ion microscopes have Angstrom-level resolution with good contrast and better charge-compensation compared to a FE-SEM and have been tested on plant samples [24]. To date, the author is not aware of any publications using a cryo-He-ion microscope; however, one instrument is currently under construction at the University of Cambridge, UK.

## 5. CryoSEM Applications in Plant Science

Examples where cryoSEM has given new information as part of original research is described in the following subject sections.

### 5.1. Plant–Microbe Interactions

Cryo-SEM-based techniques are invaluable for exploring the interactions between symbiont fungi and plant root systems, in particular, the arbuscule-forming fungi that comprise the arbuscular mycorrhiza. Colonised root samples are plunge-frozen in liquid N and then cryo-planed to give a flat imaging surface [25,26] or fractured to give a rough surface [27,28]. The frozen flat or fractured surfaces of root cortical cells are then sublimated. This sublimation step is key for defining the hyphae between cells and the variable degrees of branching and maturation of the arbuscules inside the cells [25,27]. Morphology and subcellular organisation by the electron detectors have been complimented by X-ray microanalysis that identified key differences in elemental nutrient content between the various fungal structures [25,26]. In one study [26], the authors noted that, despite the relatively slow freezing rates, there appear to be no obvious defects at the cell- and large-subcellular compartment level.

### 5.2. Plant Anatomy, Patterning and Tissue Organisation

Observations of cell arrangements within a tissue commonly require low magnifications and, in many situations, these might be met by relatively inexpensive light (stereo)microscopes and modern digital microscopes. Observing cells, whether loosely attached or well packed within a tissue, requires clear demarcation of cell boundaries and can be met by various staining techniques, fluorescent markers or light contrasting techniques such as Differential Interference Contrast (DIC). Some cell types, however, are better suited to cryoSEM for “macro” scale imaging. Root hairs are surprisingly difficult to observe and are easily damaged and commonly collapse or deform upon drying. Systematic imaging of root hairs from where they emerge from the root epidermis is amenable to cryoSEM, providing care is taken when first mounting the root on the cryo-stub. Locations and measurements of epidermal cell types: the trichoblasts and atrichoblasts can be easily compared between genotypes [29]. As an example of a tip-growing cell-type, defects in root hair cell elongation, bulging and bursting have been characterised [30].

There is clearly an added value in obtaining surface information of a plant cell, including any small changes in shape and minor deformations and surface protrusions which can be missed with light microscopy. The complex topography of some fern antheridia demonstrates the need for imaging the native, hydrated tissue [31], while detailed images of *Vittaria graminifolia* gemmae and associated meristems have allowed the various developmental stages to be pieced together to produce a time-resolved body plan of gemma formation right through to a pre-programmed pattern of abscission [32]. The dataset clearly shows the oriented divisions, the varying cell morphology and abscission scars of these elaborate structures (Figure 4a–c).

When looking at the leaf surfaces of some alpine plant species, we are presented with several challenges when using light microscopy. There are no protocols for introducing live fluorescent reporters and a very thick cuticle and outer cell wall limits the uptake of fluorescent dyes. They may also have shiny surface regions (wax, surface crust and reflective hairs) that give glare with epi-illumination. These types of plants have worked very well for cryoSEM, and their habitation in cold, dry climates appears to greatly limit the amount of exogenous (surface) ice that forms during plunge-freezing [35,36,37].

Epidermal plant tissues vary greatly in terms of cell morphology, surface roughness and overall profile—whether it is a flat surface (e.g., leaf surface), cambered (shoot meristem) or highly curved (leaf margin). Once frozen and maintained in the SEM chamber, various imaging modalities can be employed to gain different information. A secondary electron detector gives topological information--of the surface and, at high kV, a backscattered electron detector is particularly suitable for detecting cell outlines. Three dimensional information can be calculated by acquiring images of the specimen at different angles (stereophotogrammetry) or taking separate images from each quadrant from a 4Q-BSD detector [38].

### 5.3. Fluid- and Air-Filled Spaces

It is reasonable to assume that the rapid freezing at the heart of cryoSEM sample preparation not only preserves tissue integrity but also preserves the dimensions of any spaces or volumes. A cryofracture through a turgid cell reveals a wall outline filled with the more fluid cellular contents with no obvious gaps within this outline. There are the air spaces that can exist between cells, from small ones to vast areas between distant cells, as frequently observed in the leaf mesophyll (Figure 4d). This does require careful interpretation as the cryofracture can remove loosely attached cells, resulting in an enlarged space. Extracting quantitative data on mesophyll intercellular space as a function of tissue or whole leaf area provides useful physiological information. For example, cryoSEM images of leaf cryofractures have found marked differences in intercellular dimensions as a function of excess zinc [39] and during leaf development [40], which may be linked to photosynthetic fitness. The connection of these intercellular spaces with the leaf exterior occurs via the size-regulated pores known as stomata, and cryoSEM allows for a preserved snapshot for accurate measurement of stomatal aperture sizes [41].

Stomata and the air-filled voids between leaf mesophyll are at the motor end of transpiration and are key conduits that generate the strong negative pressure for the upwards flow of xylem sap through otherwise empty vessels. The sap has long been thought of as a continuous column from the root, through the shoot, with continuous flow interrupted by air-filled cavities (cavitation process) producing embolisms. In the 1990s, cryoSEM techniques were applied to the observations of sap-filled xylem vessels and the formation of cavities (e.g., see [42,43]). The high frequency of cavitation challenged the long-standing assumption of the mechanism describing upwards water transport, with new cryoSEM data suggesting extensive cavitation on a predictable daytime schedule. However, follow-up work by Cochard et al. unearthed a possible artefact: the freezing of the sap, specifically when it is under negative pressure, generates cavities during suboptimal cooling rates [44]. Sap-filled vessels without cavities were well preserved at raised water potential, making cryoSEM the key factor of embolism artefacts on low water potential samples. It is presently unknown whether better cooling rates using modern techniques could remedy this.

CryoSEM has allowed simple discrimination of xylem vessel luminal contents in earlywood of *Quercus serrata* [45]. The authors observed, after prolonged sublimation, that cytoplasm exhibits “resistance” to sublimation (non-sublimable), and therefore remains clearly visible, whereas the xylem sap retreated to give an empty lumen (sublimable). Vessels that are non-sublimable have not yet undergone terminal differentiation to produce mature conduits for the xylem sap, unlike sublimable vessels which contain sap. Applying this technique during earlywood formation found that earlywood vessels were transporting water at the top of the stem earlier than the corresponding vessels at regions further down the stem. This means that vessel maturation during earlywood formation occurred basipetally.

The above studies are restricted to observation and measurement at the fracture surface and the approach clearly becomes limited when trying to extrapolate the total volume of air space in, for example, a whole leaf or leaf disk. Here, nanometre resolution is not important, and microCT methods appear to be better suited for this purpose [46].

### 5.4. Plant Cell Wall/Apoplast and Cuticle

A key driver of plant growth and morphogenesis is the primary cell wall, a matrix of diverse polysaccharides and proteins that undergoes dynamic changes to facilitate cell expansion. Derbyshire et al. [47] used cryoSEM as a key tool to measure native cell wall thickness during Arabidopsis seedling growth, specifically hypocotyl elongation. Slush N_2_-plunge-frozen seedlings were transversely fractured and then coated, revealing fully intact walls across all cell layers and tissues. These fully hydrated, unprocessed and lightly manipulated samples represent a snapshot of growth for which numerous quantitative data can be acquired. A key set of reported measurements were (i) wall volume, calculated from individual cell wall thicknesses, (ii) cell perimeter and (iii) length of the hypocotyl where meaningful values depended upon no shrinkage or distortion of the samples. Arabidopsis seedlings are inherently a “wet” sample, where maintaining hydration is key. Holding a seedling for a short length of time out of the petri dish results in obvious drying and damage, and so the rapid mounting, plunge-freezing with no further manipulations, apart from the cryofracture and nanometre coating, makes cryoSEM particularly suitable for this type of study. The authors found that some walls could get very thin, nearing a 50 nanometre minimum—a figure that is not easily measurable with light microscopy. Furthermore, sectioned material prepared for TEM yielded higher wall thickness measurements compared to those of cryoSEM, demonstrating that wall thickness is defined by forces between cells that can only be preserved in frozen intact samples. Given that one of the key load-bearing components of the cell wall is believed to be the cellulose microfibrils, a 50 nm wall is equal to around 15 layers of tightly packed cellulose microfibrils; however, the primary wall is a nanostructured matrix with cellulose interspersed with pectins and hemicelluloses. Further applications of the technique to discriminate the microfibrils from the rest of the cell wall matrix in its native hydrated context would give further insight into the precise load-bearing contributions of cellulose.

The woody secondary cell walls that provide structural support and strength to tissues in, for example, the stem, are key for plants and trees to maintain an upward habit. This type of wall is predominantly made up of cellulose and contributes to the bulk of plant biomass. Any information, therefore, on the architecture of these walls has implications for materials and biofuels. For trees, dry/stored wood samples are readily available for conventional SEM analysis and provide high-resolution information on the end products. For fully hydrated “living” wood of growing plants and trees, cryoSEM has confirmed the presence of the cellulose macrofibril in cryofractured samples (Figure 4e and [33]). The macrofibril, in the order of 25 nm diameter, is currently the smallest resolvable unit of the woody secondary wall and its size, measured by cryoSEM imaging, is influenced by other wall components which are likely incorporated with the smaller cellulose microfibrils. These macrofibrils have been observed and measured in live material from both softwoods and hardwoods. The future challenge is to resolve the architecture within the macrofibril itself.

Observing these secondary wall macrofibrils with cryoSEM is relatively straightforward in comparison to the individual micro-/macro-fibrils that make up the primary wall of growing cell types, dominated by non-cellulose polysaccharides. To a certain extent, such observations may improve with high-end FESEMs and He-ion microscopes. Recently, cryoSEM was used to see the effect of altered pectin chemistry upon the walls of the wavy, jigsaw-puzzle-shaped cells that make up much of the cotyledon epidermis in Arabidopsis [48]. Nanoimaging, using advanced light microscopy techniques, revealed the presence of vertical columns of specific pectin epitopes. Vertical columns or “beams” were observed, by cryoSEM, along the anticlinal walls, and their thickness could be manipulated using inducible plants with altered pectin chemistry, giving rise to the “expanding beam” model of pectin-driven generation of cell lobes. This work also gives rise to a caveat for cryoSEM observations of plant cell walls: we should not assume that any observed fibrillar structures are (exclusively) cellulose.

Pectins are the major contributor to cell–cell adhesion and affect fruit ripening and spoilage. Apples that are described as “mealy” refer to the undesirable texture and dryness when biting into the fruit and are partly a result of pectin modifications. Lapsley et al. [49] compared conventional SEM with cryoSEM for apple varieties with differing texture properties. For conventional SEM, cell collapse was observed and integrity depended upon the sample preparation with best results observed for fixed tissue. Even with optimally fixed and CPD-treated tissue, cells exhibited partial collapse; however, the data were sufficient to identify fundamental differences between non-mealy (fractures through cells) and mealy tissue (fractures around cells). CryoSEM yielded generally better preservation (turgid cells) with detailed views of cell-to-cell junctions that were found to be tightly pressed together in a non-mealy sample compared to cell-separation and intercellular ruptures, giving rise to spaces between cells in the mealy tissue being consistent with a loss of firmness in the latter. An important point, however, is that it could not be determined whether the ruptures were present before the cryo-preparation procedure or whether they were a result thereof, potentially incurring cryo-fractures along the weaker middle lamellae in the mealy samples.

The outer surface of the plant epidermis consists of the cuticle and some plants possess surface epicuticular waxes. Cryo-preparation of samples revealed amorphous rather than the expected crystalline structures of the wax, and whether this was the natural state or some cryo-artefact remains uncertain. Jeffree and Sandford [50] used cryoSEM and on-stage-sublimation to observe a real-time transition from amorphous to crystalline wax and concluded that the amorphous state was in fact contaminating water ice from the atmosphere. Modern cryoSEM preparation involves no exposure to atmosphere after freezing and so one would not expect this artefact, except wet or damp plant samples that are still a source of contaminating ice. As a key tool for observing wax structures, cryoSEM has also demonstrated a link between absence of very long wax crystals and a non-glaucous variety of wheat [51].

Some high-resolution images with intricate surface detail have been achieved for characterising and locating aggregations of plasmodesmata (PD), pores that cross the apoplast to connect adjacent cells [52]. Cryofractures of tobacco leaf trichomes were carried out by either direct contact with a cold fracture knife or by placing the upper leaf surface in colloidal graphite before freezing and then stripping away the leaf tissue (and so leaving the trichomes embedded in the graphite). Different views of the pore structures correspond to different fracture faces at the cell junctions. FE-SEM imaging under cryo-conditions resolved the desmotubule, the endoplasmic reticulum connector that runs through the PD, at good level of detail for what is essentially an unstained, unfixed, small structural compartment.

### 5.5. Cell Organelles/Endomembranes

Useful information on subcellular compartments often requires high resolution and 3D contextual information. A recent review [53] gives examples of various plant endomembrane imaging methods. The authors make the point that while fluorescence-based techniques are well suited to 3D imaging, resolution is at best, with super-resolution techniques, limited to the tens of nanometres, and the highest resolution imaging is not always applicable to fresh unprocessed samples. Cryo preparation by high-pressure freezing, and then electron tomography (acquisition and processing of a volume at multiple angles), is a powerful technique but does require processing from freeze substitution through to adequate staining. Building on this technique is the use of Cryo-FIB-SEM workflows for plunge-freezing samples to then make and retrieve the lamellae for subsequent cryo-ET of the processed material. However, what about using cryo-FIB-SEM to do the imaging as well? With the additional tools that are commonplace in a cryo-FIB-SEM, such as sublimation and coating (e.g., Pt deposition post-milling), membranous structures are easily discernable in plant tissue [23]. Furthermore, iterations of milling and cryoSEM imaging have produced high-quality volumes containing complete organelles and nanogold markers, as demonstrated for HeLa cells [54]. The same iterative method would be a powerful technique for plant cells.

A key goal in the cryoSEM technique is to observe very small structures, at the protein scale, within the context of a fractured, fully hydrated and unprocessed tissue. Unlike cryo-TEM methods that principally study the structures of a purified protein or very small sample area, cryoSEM is, in principle, better suited to observing how a high molecular weight protein or multiprotein complex is distributed within and across cells. Observing large protein complexes, in the range of tens of Angstroms, in a fractured tissue is exceptionally challenging. The sample needs to be vitrified, therefore the size of the sample is restricted and requires specialist fast-freezing apparatus. The beam characteristics of the SEM, requiring a field-emission gun, need to be fully optimised and, with the high magnification (which could be significantly above 100,000×), avoiding beam damage and yielding sufficient contrast of the imaging surface become major hurdles to overcome. Charuvi et al. have achieved such cryoSEM preparation and imaging at the level of large macromolecular complexes (≥100 Angstroms e.g., photosystem II) that reside in thylakoid membranes. The membranes are not enriched in any way, but are simply cryofractured across a portion of leaf tissue [34]. The restriction on sample size is that which fits the stub of a “sandwich” that undergoes jet freezing in a high-pressure freezer. This part achieves the required vitrification. Cryofracture produces a natural imaging face that includes separation of the thylakoid membranes and at high magnification the raised spots, representing the macro-protein complexes, are just visible (Figure 4f). Further magnification has been achieved through dual thin layer Pt-C/C coating of approximately 8 nm [55] resisting beam damage at 10 kV gun voltage. At these high magnifications, the different thylakoid macromolecular complexes can be identified and quantified.

### 5.6. Phase Separated Materials

CryoSEM is commonly used to study foodstuffs where liquids and soft matter can exhibit micro-heterogenous distributions of their constituents. This can give important information on the size and distribution of the components and structural information on the interface between two phase-separated materials in, for example, a Pickering emulsion. CryoSEM is often used for detailed characterisation of emulsions such as mapping the distribution and identifying effects upon a water-oil emulsion of maize storage protein [56], corn fibre gum [57] and soy protein-anthocyanin nanospheres [58]. To maintain the microstructures of droplets in such emulsions, it has been suggested that cryo-preparation by high-pressure freezing is more suitable than plunge-freezing (immersion) methods [59].

CryoSEM characterisation of phase separation within plant cells is a useful tool for understanding cell function. This has been applied to, for example, anthocyanic vacuolar inclusions [60] and oil droplets [61] within the cell. At the cell surface (the plasma membrane), fluorescence microscopy techniques have been invaluable for studying membrane microdomains [62,63], but the contribution of low-temperature SEM imaging has not been explored. Beyond the plasma membrane, CryoSEM imaging alongside fluorescence imaging with glycan-targeted probes suggest that phase separation of some cell wall constituents does occur [48,64,65].

## 6. Future Directions

Future technology improvements could involve cryo-preparatory hardware, such as minimizing manual handling during freezing and transfer and advances in fracturing or milling. Improvements could further involve imaging within the SEM chamber. Higher magnification and resolution of smaller features will need a transition from near-vitrification to vitrification. For plant research, sample types are often large, especially compared to the common thin films and lamellae for cryo-TEM, where there will always be a trade-off between sample size and achievable preservation. It is not always possible to fit dissected plant parts in, for example, the planchette of a high-pressure freezer, and so matching the type of output, in terms of imaging data, with sample preparation, will always need careful consideration. For low (<800×)-magnification imaging at the tissue scale, recent developments aim to make cryoSEM more accessible to plant scientists by using standard equipment without requiring expensive 3rd party cryo-hardware and consumables [66].

Within the SEM, there is an increasing number of modalities. Commercial solutions are available for integrating light microscopy techniques within the chamber including fluorescence and Raman microscopy. Thin samples exposing the cell cortex are being prepared by sonicator-unroofing techniques, and these samples can be frozen and examined with both scanning electron and transmission image detectors in a S(T)EM (Scanning Transmission Electron Microscope [67]). For 3D information, iterations of ion beam milling and imaging will allow reconstruction from the tomograms, or there is promising potential for in-chamber (cryo) X-ray tomography on intact plant samples.

## Figures and Tables

**Figure 1 plants-11-01113-f001:**
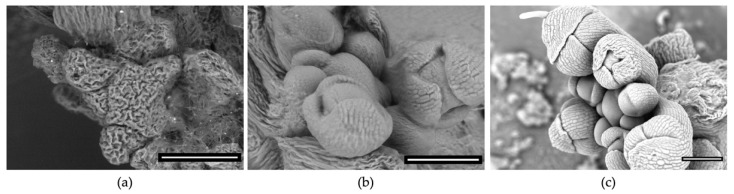
Comparison between (**a**) VP-SEM, (**b**) EP-SEM and (**c**) cryoSEM of the *Arabidopsis thaliana* shoot apex that includes the shoot apical meristem. Exposed shoot apices are sensitive to drying and are therefore dissected and prepared under water. All three images were taken using the backscattered electron detector in the same Zeiss EVO HD SEM when it was originally installed and operated as an environmental SEM (**a**,**b**) and after conversion to a cryoSEM (**c**). Under VP mode (in this example 30 Pa chamber pressure with sample mounted on a cool stage), the entire apex appears shriveled by the time the SEM is ready to take the first image. Considerable improvement is observed for EP mode (in this example 507 Pa with sample mounted on a cool stage) where a humid chamber environment can be carefully monitored and maintained. Cryo-mode (high vacuum and mounted on a cryo-stage), along with 5 nm Au/Pd coating, results in an intact sample with improved contrast compared to EP mode. Scale bars = 100 µm.

**Figure 2 plants-11-01113-f002:**
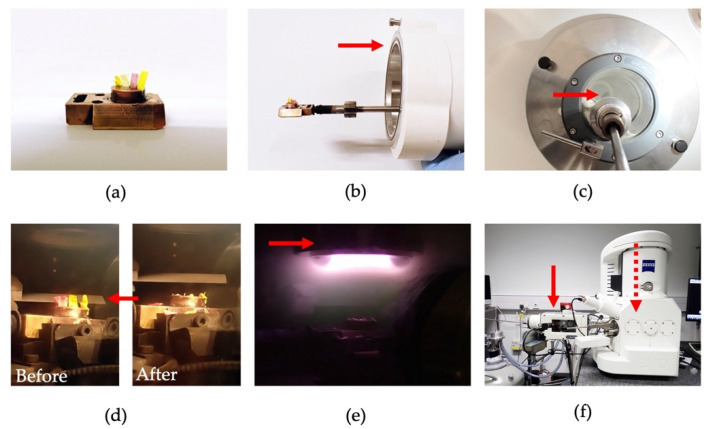
Principle steps of a cryo-preparation protocol. (**a**) Plant sample mounted on a brass cryo-stub. (**b**) Cryo-stub mounted on the transfer rod. Arrow points to the transfer capsule that will subsequently maintain the frozen sample in a vacuum. (**c**) Sample plunge frozen in slush nitrogen (arrow). Once frozen, the sample is drawn out under vacuum to the transfer capsule and then transferred to the cryo-prep stage, where it is maintained in cryo-vacuum. After the plunge-freeze, the sample is never exposed to air. (**d**) Sample on the cryo-prep stage before and after the optional fracture using the cold knife (arrow). (**e**) A sputter coater (arrow) built in to the prep deck is used to apply nanometer scale metal coatings. (**f**) The sample is then transferred from the prep-deck (solid arrow) via a passthrough port to the SEM cryo-stage (located inside the SEM chamber, dashed arrow) for imaging.

**Figure 3 plants-11-01113-f003:**
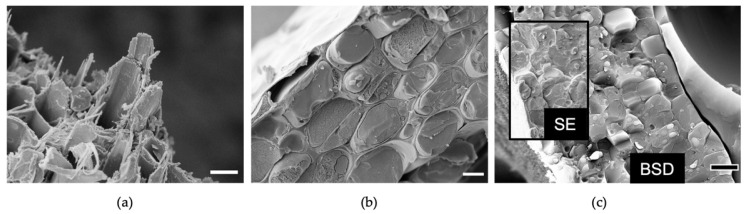
Manipulations of cryopreserved material. (**a**) Cryofracture of living *Populus tremula x tremuloides* (poplar) woody tissue showing uneven fractures of the secondary cell walls. (**b**) Sublimation of poplar stem cortex at −90 °C for 15 min. (**c**) Backscattered electron detector (BSD) image of poplar cryofracture with the SE overlay shown inset. All samples are coated with 3 nm platinum. Images acquired at 6 kV gun voltage with the SE detector and for (**c**) 25 kV with the BSD detector. Scale bars = 10 µm.

**Figure 4 plants-11-01113-f004:**
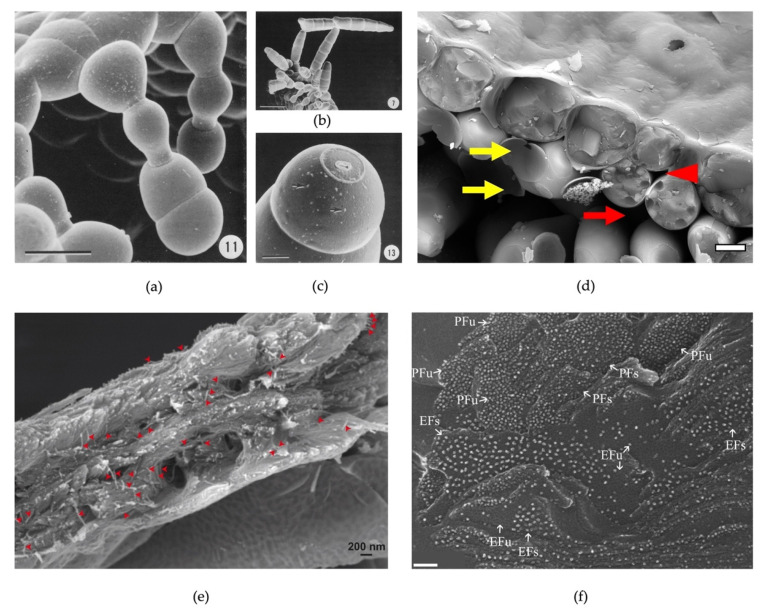
(**a**–**c**) CryoSEM of gemmae from *Vittaria graminifolia*. Note the bud scar at the end of the cell in (**c**). Scale bars = 40 µm (**a**), 100 µm (**b**), 10 µm (**c**). Figure panels taken from Sheffield and Farrar [32]. (**d**) Cryofracture of *Saxifraga lolaensis* leaf showing small (red arrowhead) and large (red arrow) air spaces between cells. Cell wall remnants of mesophyll cells can be observed (yellow arrows). Image taken using backscattered electron detector. Scale bar = 10 µm. (**e**) Cryofracture showing cellulose-containing macrofibrils (small red arrowheads) in the *Arabidopsis thaliana* secondary cell wall. Image taken from Lyczakowski et al. [33]. (**f**) High-magnification cryoSEM of fractured thylakoid membrane showing large protein complexes. Different membrane fracture faces (EF, Exoplasmic Face and PF, Protoplasmic Face) are observed in the same field of view and these are either in stacked (EFs, PFs) or unstacked (EFu, Pfu) thylakoid regions. Scale bar = 100 nm. Figure panel taken from Charuvi et al. [34].

## Data Availability

Not applicable.

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
