# Peer review of "An Overview of Cryo-Scanning Electron Microscopy Techniques for Plant Imaging"

_plants, 2022, doi:10.3390/plants11091113_

Round 1
Reviewer 1 Report
Review of the manuscript entitled : ‘An overview of cryo-scanning electron microscopy techniques for plant imaging ‘ by Raymong Wightman submitted to the Plants. Thank you very much for this set of information, quite hard to find anywhere else in a such condense form. The main gain of this study is to provide accurate information for the quality assessment and application of cryo-SEM techniques for plant cells/tissue/organs. The topic is very interesting and useful, I am convinced that this paper will be highly cited. However, the author attempts to provide data and facts without the image=proof from plant tissue. It would be great if cryo-SEM images of delicate structures will be provided, ideally in comparison with the figures from conventional SEM. For example, leaf epidermis is usually deformed, especially its emergences like trichomes, stomata, cuticle. Similar for secondary cell wall thickenings, pollen morphology or root hairs. If it is possible, could you please provide any images to encourage the readers and scientists to use the cryo-SEM against conventional SEM microscopy? It would strengthen the paper.
I think the ms is well written, it provides a lot of new information, advantage listing, useful for people dealing with micropreparation and bio-imaging.
Author Response
Reviewer 1.
Review of the manuscript entitled : ‘An overview of cryo-scanning electron microscopy techniques for plant imaging ‘ by Raymong Wightman submitted to the Plants. Thank you very much for this set of information, quite hard to find anywhere else in a such condense form. The main gain of this study is to provide accurate information for the quality assessment and application of cryo-SEM techniques for plant cells/tissue/organs. The topic is very interesting and useful, I am convinced that this paper will be highly cited. However, the author attempts to provide data and facts without the image=proof from plant tissue. It would be great if cryo-SEM images of delicate structures will be provided, ideally in comparison with the figures from conventional SEM. For example, leaf epidermis is usually deformed, especially its emergences like trichomes, stomata, cuticle. Similar for secondary cell wall thickenings, pollen morphology or root hairs. If it is possible, could you please provide any images to encourage the readers and scientists to use the cryo-SEM against conventional SEM microscopy? It would strengthen the paper.
I think the ms is well written, it provides a lot of new information, advantage listing, useful for people dealing with micropreparation and bio-imaging.
Thank you for the feedback. I did want to include more images from published original research to illustrate the advantages of cryoSEM, however, many publishers make it expensive and difficult to get the necessary permissions. What I do have are images of our most delicate structure - the shoot apex and meristem - where I have representative images from the same microscope before and after its cryo- conversion. This shows for really delicate samples that cryo is an improvement over VP and some better quality images over EP.
Reviewer 2 Report
I am not a SEM practitioner myself, but I found this review informative and pleasant to read. The author did a great job explaining the necessity of imaging fragile/hydrated plant samples under cryo conditions, the steps and setups involved, and provided a comprehensive survey of published work using cryoSEM.
Below are a few suggestions/questions.
1. It would be nice to mark the cryo-stage as well for Figure 1f.
2. Where is the cryo-He-ion microscope under construction? It would be nice to add a source/link in line 246.
3. The rapid freezing should preserve the dimensions quite truthfully, but is it possible that the air-filled space observed in Figure 3 introduced by the fracture process? I would appreciate if the author can comment on this.
4. Related to Figure 3, I cannot detect any "red arrowheads" for panel b.
5. The legend of Figure 4 needs to include the description for "PFu", "EFs", and "PFs."
6. Section 5 may be benefited from a re-organization. My suggestion is to put 5.5 in the beginning, followed by 5.4 cell organelle, 5.4 phase separated materials, 5.2, 5.3, and 5.1. This order goes from global to local and plant alone to plant-microbe interactions.
Author Response
Reviewer 2.
Thank you for the useful comments. Responses are below.
1. It would be nice to mark the cryo-stage as well for Figure 1f.
Done. A dashed line shows the location of the cryostage in the SEM chamber. Note this figure is now Figure 2f.
2. Where is the cryo-He-ion microscope under construction? It would be nice to add a source/link in line 246.
It’s being built at Cambridge and involves the author. The sentence (now line 263) has been updated giving the location of the prototype..
3. The rapid freezing should preserve the dimensions quite truthfully, but is it possible that the air-filled space observed in Figure 3 introduced by the fracture process? I would appreciate if the author can comment on this.
This is a good point. Air spaces are well preserved in cryo, however, the nature of the cryofracture means that loosely attached cells (such as leaf mesophyll) can be sometimes removed at the fracture plane. In these circumstances the site of the cell leaves behind remnants such as a partial cell wall, broken cell-cell connections or an imprint of where the cell used to press on to its neighbour. In the example in the previous version of Figure 3a, the large arrow may be pointing to a place where there was a cell and some cell wall may have been left behind. This arrow has been moved in an updated Figure 3a (now Figure 4d) to another large airspace and the remnants of removed cells highlighted. A sentence highlighting the above caveat is now included at line 342 and the figure 3 legend updated.
4. Related to Figure 3, I cannot detect any "red arrowheads" for panel b.
Figure 3b (now Figure 4f) arrowheads are now larger along with an increased size of the image.
5. The legend of Figure 4 needs to include the description for "PFu", "EFs", and "PFs."
Figure 4 legend now includes the abbreviations and their definitions.
6. Section 5 may be benefited from a re-organization. My suggestion is to put 5.5 in the beginning, followed by 5.4 cell organelle, 5.4 phase separated materials, 5.2, 5.3, and 5.1. This order goes from global to local and plant alone to plant-microbe interactions.
I see the reviewers point that the section 5 subsections could flow and structure better. My original idea was to work through from outside to inside which broadly co-incided with going from low mag to higher mags but clearly this was lost as the manuscript was added to. To help with restructuring section 5 I have combined Figures 3 and 4 in to one large figure (Figure 4new). To maintain a kind of structure/flow I have now ordered as follows:
Plant Microbe interactions
Plant anatomy, patterning, tissues
Fluid filled air spaces
Plant cell wall and cuticle
Cell organelles
Phase separation
(Broadly - Outside plant to inside plant to outside cell to inside cell AND Macro level to micro level to nano level)